# Development and validation of clinical prediction models for mortality, functional outcome and cognitive impairment after stroke: a study protocol

Marion Fahey,[1] Anthony Rudd,[1,2] Yannick Béjot,[3] Charles Wolfe,[1,4] Abdel Douiri[1,4]

¹Division of Health and Social Care Research, King's College London, London, UK
²National Institute for Health Research (NIHR) Biomedical Research Centre, Guy's and St Thomas' NHS Foundation Trust, London, UK
³Registre Dijonnais des AVC, EA4184, Dijon, France
⁴National Institute for Health Research (NIHR) Collaboration for Leadership in Applied Health Research and Care (CLAHRC), South London at King's College Hospital NHS Foundation Trust, London, UK

**Correspondence to**
Marion Fahey;
marion.fahey@kcl.ac.uk

## ABSTRACT

**Introduction** Stroke is a leading cause of adult disability and death worldwide. The neurological impairments associated with stroke prevent patients from performing basic daily activities and have enormous impact on families and caregivers. Practical and accurate tools to assist in predicting outcome after stroke at patient level can provide significant aid for patient management. Furthermore, prediction models of this kind can be useful for clinical research, health economics, policymaking and clinical decision support.

**Methods** 2869 patients with first-ever stroke from South London Stroke Register (SLSR) (1995–2004) will be included in the development cohort. We will use information captured after baseline to construct multilevel models and a Cox proportional hazard model to predict cognitive impairment, functional outcome and mortality up to 5 years after stroke. Repeated random subsampling validation (Monte Carlo cross-validation) will be evaluated in model development. Data from participants recruited to the stroke register (2005–2014) will be used for temporal validation of the models. Data from participants recruited to the Dijon Stroke Register (1985–2015) will be used for external validation of the models. Discrimination, calibration and clinical utility of the models will be presented.

**Ethics** Patients, or for patients who cannot consent their relatives, gave written informed consent to participate in stroke-related studies within the SLSR. The SLSR design was approved by the ethics committees of Guy's and St Thomas' NHS Foundation Trust, Kings College Hospital, Queens Square and Westminster Hospitals (London). The Dijon Stroke Registry was approved by the Comité National des Registres and the InVS and has authorisation of the Commission Nationale de l'Informatique et des Libertés.

## Strengths and limitations of this study

► First prognostic tool in stroke to follow, a priori, the Prognosis Research Strategy (PROGRESS) framework, and Transparent Reporting of a multivariable prediction model for Individual Prognosis Or Diagnosis (TRIPOD) reporting guidelines for prognostic research.

► Specifies statistical analysis plan and informative levels of tool performance to increase transparency of results and the final report.

► The proposed study predicts outcomes longitudinally, which may be more reflective of clinical needs than predictions made at predefined time points.

► The proposed study is restricted by the use of predictor variables measured in previous data sets and the limitations of these measures. The external validation is not independent.

and/or cognitive impairments that require ongoing active assessment and management.[2] Stroke can be seen as a chronic condition, spanning not only the incident event and formal rehabilitation but the rest of the patient's life. Rehabilitation from stroke requires a sustained, coordinated effort from informed multidisciplinary teams (MDTs), as well as patients and carers, both in the clinical setting and in the community.[3] MDTs and patients make numerous decisions on the basis of an estimated probability that a specific event will occur in the future. These predictions are used for planning lifestyle or therapeutic decisions on the basis of the risk of developing a particular outcome or state of health.[4–6] More recently such estimates are used to risk-stratify participants in therapeutic intervention trials and case-mix classifications.[7 8]

Information from a single predictor is often insufficient to provide reliable estimates of prognostic probabilities or risks, particularly

## INTRODUCTION

Stroke is one of the most common causes of serious adult physical disability and the third most common cause of death worldwide.[1] Despite the introduction of effective treatments for acute stroke, early rehabilitation and secondary prevention, the majority of stroke survivors have medical comorbidities, physical

BMJ

in complex patients, for example those with comorbidities.[9 10] Therefore probability estimates are commonly based on combining information from multiple predictors to form a multivariable clinical prediction model (CPM).[11 12]

CPMs are abundant in prognostic research literature, but few are implemented or used in routine clinical practice.[13] One explanation for this is that, although many models have been developed,[14 15] they have limited utility for clinical applications, particularly in a long-term care setting. Progression or regression of disease is highly variable both over time and between individuals.[16] Current CPMs typically estimate risk at predefined time points. With variability in mind, time series data and methods may be more appropriate for accurately capturing recovery, particularly when the aim is planning immediate and long-term care simultaneously for individual patients.

There is generally a lack of confidence among clinicians in applying risk scores in practice. Many believe there is the lack of sufficient evidence to demonstrate the reproducibility and transportability of the model in a different population.[17] To be considered useful, a risk score should be clinically credible, accurate (well calibrated with good discriminative ability) and have generality (be externally validated).[17]

We have evaluated the accuracy of existing models in predicting stroke outcomes by systematic review of literature and meta-analysis. The systematic review and meta-analysis concluded that existing models have much potential and advised to build on previous work, as opposed to designing new models, in developing CPMs suitable for the long-term care setting.[15] Models predicting outcomes longitudinally, as opposed to predefined time points, may best support patient management and so we will build on work by colleagues in this area.[18 19]

## OBJECTIVES
### Primary objectives
The primary objectives were to develop and internally validate prediction models in patients with ischaemic stroke for functional outcome, survival and cognitive impairment at 5 years, and to externally validate and update these models through an external data set of patients with ischaemic stroke.

### Secondary objectives
The secondary objectives were to assess deviations of predicted recovery curves, to investigate why patients with stroke regain different recovery that plateaued at different levels, and to derive and validate scoring system for classifying risks (case-mix classification) from these patient-centred predictive models.

## METHODS
This cohort study is informed by the Prognosis Research Strategy (PROGRESS) framework and recommendations

by authors in the field. We will report the study protocol using the Transparent Reporting of a multivariable prediction model for Individual Prognosis Or Diagnosis (TRIPOD) statement for prediction studies.[20]

### Source of data
The study is a prospective longitudinal cohort study. The data source is the South London Stroke Register (SLSR), an ongoing, prospective, population-based stroke register set up in January 1995 recording all first-ever strokes in patients of all ages for an inner area of South London. The methods of the SLSR have been described previously in detail[2] and are summarised here. Stroke was defined according to the WHO criteria and Oxfordshire Community Stroke Project Classification. Multiple overlapping sources of information are employed. Data from 1995 to 2016 will be used for model derivation and internal validation.

### Participants
Patients admitted to hospitals serving the study area (two teaching hospitals within and three hospitals outside the study area) are identified by regular reviews of acute wards admitting patients with stroke, and national data on patients admitted to any hospital in England and Wales with a diagnosis of stroke are screened for additional patients. All general practitioners (n=699 (2011)) within and on the borders of the study area are contacted regularly and asked to notify the SLSR of patients with stroke. Referral of non-hospitalised patients with stroke to a neurovascular outpatient clinic (from 2003) or domiciliary visit to patients by the study team is also available to general practitioners. Community therapists are contacted every 3 months. Death certificates are checked regularly. Patients are assessed at the stroke onset, 3 months and annually after stroke. Participants receive treatments in line with the UK national clinical guidelines for stroke. Completeness of case ascertainment has been estimated at 88% by a multinomial logit capture–recapture model using the methods described in detail elsewhere.[21]

### Outcome
The outcomes of interest are mortality measured as time to event, functional outcome measured using the Barthel Index (BI) and cognitive impairment measured using the Mini-Mental State Exam (MMSE) or Abbreviated Mental Test (AMT), measured at baseline, 3 months and annually up to 5 years after stroke. Specially trained nurses and field workers collect all data prospectively whenever feasible. A study doctor verifies the diagnosis of stroke. Patients are examined within 48 hours of referral to the SLSR where possible. Follow-up data are collected by validated postal or face-to-face instruments with patients and/or their carers. Outcome definition and measurement method are the same for all patients. Data collection is carried out by a third party, uninvolved in this study.

### Predictors
Candidate predictors to be considered in the prediction models will be based on predictor variables

as identified in previous systematic reviews in this field,[15 22] ease and reliability of measurement in clinical setting and theoretical association with the progression of outcomes. The number of variables required to ensure adequate power based on our target sample size (see below), to avoid overfitting and to encourage parsimony and applicability of the model in clinical practice will also be considered. Number, type, definition, method for measurement and handling of candidate predictors in the modelling are listed in the data supplement. All predictors are measured on patient presentation. Data collection is carried out by a third party, not involved in this research.

## Sample size

Cumulative survival up to 10 years after stroke is estimated at 63.7%, 42.8% and 24.0%, surviving up to 1, 5 and 10 years, respectively.[2] Disability is estimated on average at 11% (10-year average and standardised to European population in the SLSR).[2] Current SLSR data (1995–2015) estimate disability (BI >15) at 24.6%, 23.7% and 26.3% at 1, 5 and 10 years, respectively. Cognitive impairment (MMSE <24) is estimated at approximately 22% in the first 5 years following stroke (age-standardised to European population).[23] Current SLSR data (1995–2015) estimate cognitive impairment at 29.8%, 28.9%, 28.6% at 1, 5 and 10 years, respectively. Rules of thumb for fitting multivariate models suggest that 10 events for every variable (EPV) are required to avoid overfitting in model development studies. Although we are within these limitations, the use of this rule can result in small sample sizes, which may lead to overfitting and optimism. It is recommended that EPV should be data-driven.[24] Sample size calculations described by Jinks[25] will be used for survival analysis. Simulation will be used to determine appropriate sample size for prediction models for functional outcome and cognitive impairment.

## Missing data

Missing data are inevitable in studies with long follow-up and may lead to bias and imprecision. Multiple imputation will be used to impute missing values, under a missing at random assumption, so as to reduce bias and avoid excluding participants from the analysis.[26–30] Imputations typically break down when missing data are excessive; therefore, data with more than 80% missing data will be excluded.

## Statistical analysis methods

We will develop three CPMs for the outcomes: (1) mortality, (2) functional outcome and (3) cognitive impairment up to 5 years after stroke. The start point is time of stroke and end point is 5 years post stroke.

For predictor selection during multivariate modelling, a variable selection and shrinkage procedure will be used to decide which of the identified candidate predictor variables should be included in the final prediction model. Continuous variables will be kept as continuous (rather than say dichotomising) to avoid loss of power. Non-linear trends will be considered using fractional polynomials and the multivariable fractional polynomial procedure. Clinically meaningful interactions (eg, time) will be included in the model.

We will assess internal validity with a bootstrapping procedure for a realistic estimate of the performance of prediction models in similar future patients.[5 12] The bootstrap validation approach uses all of the data to develop the prediction model and provides a mechanism to account for model overfitting or uncertainty in the entire model development process, thereby quantifying any optimism in the final prediction model. Also, it provides for estimating a shrinkage factor that can be used to adjust the regression coefficients and apparent performance for optimism, such that better performance will be obtained in subsequent model validation studies and applications.[31]

External validity will be assessed using data form the Dijon Stroke Register (DSR). Patients will be classified using the estimates of the previously developed models and performance assessed. Subgroup analyses will be carried out to assess deviations from predicted recovery trajectories and investigate why patients with stroke regain different recovery that plateaued at different levels.

## Risk groups

Although risk groups (eg, 'high risk', 'moderate risk', 'low risk') may make models more accessible, no risk groups will be created. There is no clear consensus on how to create risk groups or how many groups to use.[31] There are concerns that use of risk groups may not be in the best interest of patients and may become standard, although lacking any rational.[31] Also, the simplification of predicted probabilities assumes risk is the same for all individuals within that category.

## Development versus validation

Data from the DSR will be used for external validation; its methods have been described previously in detail[32–34] and both data sets are contrasted here. The DSR and SLSR are population-based registers; multiple overlapping sources of notification are used and stroke is defined according to WHO criteria for both. Specially trained field workers collect all data. This includes sociodemographic factors, disease characteristics, patient history and cardiovascular risk factors in both registers. DSR participants are followed up at time of stroke and yearly thereafter; these questionnaires are administered in outpatient clinics or conducted telephonically by clinical nurses. Follow-up procedures in SLSR are similar, but participants are followed up at 3 months also. In both registers, survival is measured using national data, cognitive status is measured using either the AMT or the MMSE, and disability is measured using BI.

## REPORTING OF RESULTS
### Participants

The flow of participants through the study, including the number of participants with and without the outcome, and a summary of the follow-up time will be described. The characteristics of the participants, including the number of participants with missing data for predictors and outcome for both development and validation cohorts, will be provided.

### Model development

The number of participants and outcome events in each analysis will be presented, as well as the unadjusted association between each candidate predictor and outcome.

### Model specification

The full prediction model for each outcome will be presented, including all regression coefficients, and model intercept and baseline survival. Once a final model is identified, methods will be applied to simplify and adapt the presentation of the model to a scoring system to facilitate its application in practice at a later date. An explanation of how to use the model and scoring system will also be presented.

### Model performance

Model performance assessment has been designed using the framework described by Steyerberg and colleagues.[35] Model performance will be assessed in derivation and validation data sets. For model development studies, we are primarily interested in discrimination, because the model will be well calibrated (on average) by definition when smoothing methods are used. In validation studies, assessment of both discrimination and calibration is fundamental.[35 36] Calibration of the derived models will be measured using calibration plot and Hosmer-Lemeshow test or counterpart test for the survival model.[37 38] Furthermore, they convey no indication of magnitude or direction of any miscalibration; hence, calibration plots will also be presented. Calibration plots will also be evaluated in relation to key predictors/subgroups. Discrimination of the derived models will be measured using the concordance statistic and CIs (c-statistic). The c-index is identical to the area under the receiver operating characteristic curve for models with binary end points and can be generalised for time-to-event (survival) models accounting for censoring. A benchmark level of discrimination was determined in our meta-analysis for mortality and functional outcome. On the basis of the reference standard values in statistics literature[39] and those of previous work in stroke, we will consider area under the curve values of greater than 0.8 to be acceptable for these outcomes. Classification measures (eg, sensitivity, specificity, predictive values, net reclassification improvement) will be presented and cut points selected a priori. Decision curve analysis will be undertaken to assess clinical utility.[40]

### Model updating

We will update the model if it shows poor performance in external data (DSR) by recalibration or revision methods depending on discrimination performance (c-statistics <0.80). If model is updated, updating approaches recommended by Steyerberg et al[13] (model recalibration, model revision, model extension) will be adopted as appropriate.

## CONCLUSION

We have described the methods and statistical analysis plan to develop and to validate a family of CPMs for stroke over the long term. To our knowledge, this tool will be the first of its kind in stroke to follow, a priori, the PROGRESS framework and TRIPOD reporting guidelines for prognostic research. Importantly for secondary prevention, the tool will be developed specifically to predict the progression of disease and to identify those at high risk of an adverse outcome. Results coming from this study will be interpreted for both clinical and research purposes.

**Contributors** MF: drafting of article. AR: critical revising of the article. YB: critical revising of the article. CW: critical revising of the article. AD: critical revising of the article.

**Competing interests** None declared.

**Ethics approval** The SLSR design was approved by the ethics committees of Guys and St Thomas NHS Foundation Trust, Kings College Hospital, Queens Square and Westminster Hospitals (London). The Dijon Stroke Registry was approved by the Comit National des Registres and the InVS and has authorisation of the Commission Nationale de lInformatique et des Liberts.

**Provenance and peer review** Not commissioned; externally peer reviewed.

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
