## [Reviewer comments · BMJ Open]

ARTICLE DETAILS

TITLE (PROVISIONAL)	DEVELOPMENT AND VALIDATION OF CLINICAL PREDICTION MODELS FOR MORTALITY, FUNCTIONAL OUTCOME AND COGNITIVE IMPAIRMENT AFTER STROKE: A STUDY PROTOCOL
AUTHORS	Fahey, Marion; Rudd, Anthony; Béjot, Yannick; Wolfe, Charles; Douiri, Abdel

VERSION 1 - REVIEW

REVIEWER	Martin Dennis University of Edinburgh United Kingdom I have a longstanding interest in the development and uses of predictive models in stroke and have received grant funding to develop predictive models. I am supervising some research with similar objectives.
REVIEW RETURNED	20-Oct-2016

GENERAL COMMENTS	In the abstract the authors say "We will construct multilevel models and Cox proportional hazard model to predict recovery trajectories for cognitive impairment, functional outcome and mortality up to 5 years after stroke. " To predict a trajectory sounds a bit different from predicting an outcome at different time points - I think they are planning to predict outcome at specific time points rather than the shape of a recovery curve? I am not entirely clear whether information captured after baseline will be incorporated into the models to refine the predictions
--

REVIEWER	Amy Brodtmann University of Melbourne Australia
REVIEW RETURNED	31-Oct-2016

GENERAL COMMENTS	The authors present a prognostic model for the prediction of cognition decline and impairment following stroke, using retrospective data from the South London Stroke Register (two 10 year periods, derivation and internal validation sample 1995-2004, temporal validation is from participants from 2005-2014, 2869 patients). They will validate the model using further retrospective data from the Dijon Stroke Register (30 years, 1985-2015, 2172 patients). They aim to develop models to "predict recovery trajectories for cognitive impairment, functional outcome and mortality up to 5 years after stroke". These are ambitious and require impeccable longitudinal data on all subjects.
--

	Strengths: Study design: excellent cohorts giving large sample size and ability to validate in comparative group. Weaknesses: The Dijon group presents a sample over 30 years with likely changes in measures impeding seamless comparisons; poorly written; concerns about modelling of cognitive impairment given the paucity of information especially on how this was measured. I have a number of questions and comments for the authors:  1. The paper is quite clumsily written with grammatical errors and missing words (e.g., “informed multidisciplinary (MDTs)” should read multidisciplinary teams?; no commas after authors in references; no full-stop after et al.; no spaces between brackets and text, e.g., page 5, lines 42-43, “Kamper et al and Royston et al (Kamper et al 2011; Royston et al 2009).To”; missing text, e.g., page 6, line 23 “admitting stroke patients, , national data”, etc.). Some sections appear clearly cut-and-pasted, even having different fonts (e.g., page 4, line 44 to page 5, line 3), and others still have track-changes formatting evident (e.g., page 7, line 4). This sometimes makes it difficult to follow the flow of logic and appear haphazardly formatted, which does not make for an easy read. 2. Please describe how patients are assessed at the 3 month and annual check-ups. Are these telephone interviews or face-to-face? Is an NIHSS done, or Rankin, or any cognitive screening? 3. How was disability assessed? Is this only on the Bartel Index? I know you reference other papers of previous work, but in order for the reader to understand your methods a brief overview is required. 4. Page 8, lines 35-36: “Current SLSR data (1995-2015) estimates cognitive impairment at 29.8%, 28.9%, 28.6% at 1, 5 and 10 years respectively.” How was this estimated? What cognitive screen was used? Did all participants receive the same testing? 5. One of the three aims is to model/predict cognitive impairment, but there is scant information on how this was assessed. This is vital for our understanding. There needs to be a section on the definition of cognitive impairment for the purposes of this study, what cut-offs were used, whether functional decline was also assessed, and what tools were used. This is particularly important for the validation component, as you will be comparing different language tools.
--	--

REVIEWER	Thomas Liman Center for Stroke Research Berlin Charité Universitaetsmedizin Berlin Berlin, Germany
REVIEW RETURNED	31-Oct-2016

GENERAL COMMENTS	This is an interesting study protocol for the development and validation of predictions models for long-term outcome after stroke from the well-known team from the South London Stroke Register. In summary, prognostic models should be developed with advanced statistical methods In 2869 patients with first stroke from the London population-based stroke register and validated in the well-known Dijon stroke register. The paper is well written and methods are appropriate and up-to-date. As stated by the authors current model and prognostic scores in stroke medicine are limited and not useful for clinical practise. Thus, in my opinion to publish this protocol and methodology is of great importance for other stroke researcher.
--

	There are only some (minor) points that I like to suggest:  • Limitations of MMSE/AMT in detecting cognitive impairment should be stated • Method of follow-up examination, data ascertainment etc. (telephone, letter, outpatient clinic, questionnaires) should be explained in more detail.
--	---

VERSION 1 – AUTHOR RESPONSE

Reviewer: 1

In the abstract the authors say "We will construct multilevel models and Cox proportional hazard model to predict recovery trajectories for cognitive impairment, functional outcome and mortality up to 5 years after stroke. " To predict a trajectory sounds a bit different from predicting an outcome at different time points - I think they are planning to predict outcome at specific time points rather than the shape of a recovery curve? I am not entirely clear whether information captured after baseline will be incorporated into the models to refine the predictions

Updated text in abstract:

"We will use information captured after baseline to construct multilevel models and Cox proportional hazard model to predict recovery for cognitive impairment, functional outcome and mortality up to 5 years after stroke."

Reviewer: 2

The authors present a prognostic model for the prediction of cognition decline and impairment following stroke, using retrospective data from the South London Stroke Register (two 10 year periods, derivation and internal validation sample 1995-2004, temporal validation is from participants from 2005-2014, 2869 patients). They will validate the model using further retrospective data from the Dijon Stroke Register (30 years, 1985-2015, 2172 patients). They aim to develop models to "predict recovery trajectories for cognitive impairment, functional outcome and mortality up to 5 years after stroke". These are ambitious and require impeccable longitudinal data on all subjects.

Updated text in "missing data":

"Missing data is inevitable in studies with such long follow up and may lead to bias and imprecision. Multiple imputation will be used to impute missing values, under a missing at random assumption, so as to reduce bias and avoid excluding patients from the analysis (Moons et al 2012..."

Strengths:

Study design: excellent cohorts giving large sample size and ability to validate in comparative group. No edits required

Weaknesses:

The Dijon group presents a sample over 30 years with likely changes in measures impeding seamless comparisons; poorly written; concerns about modelling of cognitive impairment given the paucity of information especially on how this was measured.

A paragraph entitled "Development vs. Validation" has been added to the manuscript. This describes differences between SLSR and DSR in setting, eligibility criteria, outcome and predictors.

Examples highlighted corrected and post-revisions manuscript sent to professional proof reading service.

Outcomes have also been critically discussed in more detail in the "outcomes" paragraph.

I have a number of questions and comments for the authors:

1. The paper is quite clumsily written with grammatical errors and missing words (e.g., “informed multidisciplinary (MDTs)” should read multidisciplinary teams?; no commas after authors in references; no full-stop after et al.; no spaces between brackets and text, e.g., page 5, lines 42-43, “Kamper et al and Royston et al (Kamper et al 2011; Royston et al 2009).To”; missing text, e.g., page 6, line 23 “admitting stroke patients, , national data”, etc.). Some sections appear clearly cut-and-pasted, even having different fonts (e.g., page 4, line 44 to page 5, line 3), and others still have track-changes formatting evident (e.g., page 7, line 4). This sometimes makes it difficult to follow the flow of logic and appear haphazardly formatted, which does not make for an easy read.

Examples highlighted corrected and post-revisions manuscript sent to professional proof reading service.

2. Please describe how patients are assessed at the 3 month and annual check-ups. Are these telephone interviews or face-to-face? Is an NIHSS done, or Rankin, or any cognitive screening?

Updated text:

“ Specially trained nurses and field workers collect all data prospectively whenever feasible. A study doctor verified the diagnosis of stroke. Patients are examined within 48hours of referral to the SLSR where possible. Follow up data were collected by validated postal or face-to-face instruments with patients and or their carers.”

3. How was disability assessed? Is this only on the Bartel Index? I know you reference other papers of previous work, but in order for the reader to understand your methods a brief overview is required.

Updated text:

“The outcomes of interest are Mortality, measured as time to event, functional outcome, measured using the Bartel Index, and Cognitive Impairment, measured using the (Mini Mental State Exam or /Abbreviated Mental Test). Outcomes are measured at baseline, 3 months and annually up to 5 years after stroke”

4. Page 8, lines 35-36: “Current SLSR data (1995-2015) estimates cognitive impairment at 29.8%, 28.9%, 28.6% at 1, 5 and 10 years respectively.” How was this estimated? What cognitive screen was used? Did all participants receive the same testing?

Updated text:

“Current SLSR data (1995-2015) estimates disability(Bartel Index > 15) at 24.6%, 23.7% and 26.3% at 1, 5 and 10 years respectively. Cognitive impairment (Mini mental State Exam <24) is estimated at approximately 22% in the first five years follow stroke (age-standardised to European population, Douiri et al 2013).”

5. One of the three aims is to model/predict cognitive impairment, but there is scant information on how this was assessed. This is vital for our understanding. There needs to be a section on the definition of cognitive impairment for the purposes of this study, what cut-offs were used, whether functional decline was also assessed, and what tools were used. This is particularly important for the validation component, as you will be comparing different language tools.

A paragraph entitled “Development vs. Validation” has been added to the manuscript. This describes differences between SLSR and DSR in setting, eligibility criteria, outcome and predictors. All outcomes have also been described in more detail in the “outcomes” paragraph.

Reviewer: 3

This is an interesting study protocol for the development and validation of predictions models for long-term outcome after stroke from the well-known team from the South London Stroke Register. In summary, prognostic models should be developed with advanced statistical methods in 2869 patients with first stroke from the London population-based stroke register and validated in the well-known Dijon stroke register.

The paper is well written and methods are appropriate and up-to-date. As stated by the authors current model and prognostic scores in stroke medicine are limited and not useful for clinical practise. Thus, in my opinion to publish this protocol and methodology is of great importance for other stroke researcher.

No edits required

There are only some (minor) points that I like to suggest:

1. Limitations of MMSE/AMT in detecting cognitive impairment should be stated
All outcomes have been discussed in more detail in the “outcomes” paragraph, including limitations of measures.
2. Method of follow-up examination, data ascertainment etc. (telephone, letter, outpatient clinic, questionnaires) should be explained in more detail

Updated text:

“ Specially trained nurses and field workers collect all data prospectively whenever feasible. A study doctor verified the diagnosis of stroke. Patients are examined within 48hours of referral to the SLSR where possible. Follow up data were collected by validated postal or face-to-face instruments with patients and or their carers.”

VERSION 2 – REVIEW

REVIEWER	Amy Brodtmann Florey Institute for Neuroscience and Mental Health University of Melbourne Australia
REVIEW RETURNED	13-Feb-2017

GENERAL COMMENTS	The authors have addressed my questions in a reasonable manner. Several spelling (e.g., Bartel not Barthel), grammatical (capitalising stroke and team in the middle of the sentence) and editing (mixed numbered and author, date references) errors persist. The paper will need careful and comprehensive editing prior to publication.
--